# Cancer Prevention and Therapy of Two Types of Gap Junctional Intercellular Communication–Deficient “Cancer Stem Cell”

**DOI:** 10.3390/cancers11010087

**Published:** 2019-01-14

**Authors:** James E. Trosko

**Affiliations:** Department Pediatrics & Human Development, College of Human Medicine, Michigan State University, East Lansing, MI 48824, USA; james.trosko@hc.msu.edu

**Keywords:** connexins, gap junctions, stem cells, oncogenes, cancer stem cells

## Abstract

Early observations showed a lack of growth control and terminal differentiation with a lack of gap junctional intercellular communication (GJIC). Subsequent observations showed that epigenetic tumor promoters and activated oncogenes, which block gap junction function, provide insights into the multi-stage, multi-mechanism carcinogenic process. With the isolation of embryonic induced pluri-potent stem cells and organ-specific adult stem cells, gap junctions were linked to early development. While tumors and tumor cell lines are a heterogeneous mixture of “cancer stem cells” and “cancer non-stem cells”, the cancer stem cells seem to be of two types, namely, they express (a) no connexin genes or (b) connexin genes, but do not have functional GJIC. These observations suggest that these “cancer stem cells” originate from normal adult stem cells or from the de-differentiation or re-programming of somatic differentiated cells. This “*Concept Paper*” provides a hypothesis that “cancer stem cells” either originate from (a) organ-specific adult stem cells before the expression of the connexin genes or (b) organ-specific adult stem cells that just express gap junction genes but that the connexin proteins are rendered dysfunctional by activated oncogenes. Therefore, cancer prevention and therapeutic strategies must account for these two different types of “cancer stem cell”.

“*There is a developing speculation and assumption that agents or circumstances are promoters or are promoting if they lead to a decrease in cell–cell communication by an interference with gap junction expression. This is a serious misconception that should be abandoned*.”M. Farber [1].

“*In this light, a new perspective for therapeutic intervention by directing differentiation and restoring communications between cancer cells is certainly relevant, and the example of gap junctions is illuminating*.”J.-P. Capp [2].

## 1. Introduction: Gap Junctional Intercellular Communication Differences between Normal and Cancer Cells

One of the universal characterizations of normal cells is that they are homeostatically regulated for growth control or are “contact inhibited” [3], can differentiate if they are stem cells, are able to apoptose, and are subject to senescence. On the other hand, cancer cells are characterized by their lack of growth control, do not terminally differentiate or apoptose normally, and do not senesce, but seem to be immortal [4]. The early electrophysiological work of Loewenstein and Kanno [5], using normal and cancerous liver cells concluded, using a limited number of rat liver cancer cells, that gap junctional intercellular communication seems to be correlated with the difference between the two types of cells. Of course, as with any novel finding in science, they were unable to draw a universal or general conclusion that might be found with all cancer cells, or to identify the stage of the complex carcinogenic process to which their findings might be related. They even raised this issue, “*Do the induced changes in communication represent intermediate stages in the genesis of cancer, that is, intermediate stages in the transformation of normal into cancer cells? Does the phenomenon reflect a genetic or a purely somatic change? We have no answers as yet to these questions*”. Yet without question, the finding they made did have strong implications that this might be a universal characterization of all cancer cells. Even to this day, in the cancer field, with all the subsequent findings related to the many unanswered properties of cancer cells and the carcinogenic process, such as oncogenes and tumor suppressor genes, one rarely sees any mention of this early observation.

Clearly, at the time of that paper, a number of important observations concerning carcinogenesis were either unknown to this group (i.e., the concept of the “initiation”/”promotion”/”progression” stages of carcinogenesis [6,7,8]; the stem cell [9,10,11,12,13,14] versus the “de-differentiation” [15] hypotheses of the origin of carcinogenesis; and the isolation and characterization of human embryonic [16,17] induced pluripotent stem cells (iPSc) [18], somatic nuclear transfer stem cells [19] and human adult organ-specific stem cells [20,21,22,23]. Although, there were some early studies that showed a few conditions that could inhibit the gap junction function [24,25], it was only later that a class of chemical agents [26,27] and genes [28,29], as well as the isolation and characterization of the postulated target cells for starting the carcinogenesis processes were made. [13,30]. *The objective of this short “Concept paper” is to examine if all cancer cells have the same correlation with the gap junction or connexin genes and their functions.*

## 2. Potential Evolutionary Changes that Led to the Emergence of Multicellularity: The Role of the Connexin Gene Family

Before one can get into answering the question of the role of gap junctions in carcinogenesis, one should take a look at the origin of the first forms of life on earth. The first forms of life were characterized as single cell organisms that arose in a harsh environment (high temperatures; excessive radiation; toxic gases and no oxygen; toxic chemicals in the water; gravity; diurnal light rhythms; etc.). Their growth control was government by the limitations of nutrients and other physical factors, as well as “quorum sensing” [31]. All living organisms need energy for individual and species survival. These organisms had to acquire genes to protect and repair their genomes, so as to prevent both excessive DNA damage and mutations or alternatively, too little adaptation, so as not to be able to adapt to inevitable environmental changes [32,33].

They performed their energy needs by metabolizing glucose via glycolysis or fermentation biochemistry, an inefficient process to produce ATP. These cells could proliferate with adequate nutrients via symmetrical cell division, with only a primitive form of the phenotypic adaptive response or “differentiation”. This is seen when single cells are exposed to either excessive oxygen or a heavy metal, such as cisplatin, an anti-cancer drug (Figure 1). These examples are shown to illustrate how this no oxygen to high oxygen level transition forced evolutionary changes that led to the selection of genes for (a) stem cells that could divide either symmetrically to maintain this primitive glycolysis metabolism or asymmetrically to differentiate in high oxygen; (b) stem cell niches that maintained a low oxygen microenvironment; (c) the biochemical ability to synthesize collagen-type cell adhesion-type molecules, which can only be formed in oxygen; (d) genes that code for direct cell-cell communication between these co-joined cells; (e) new intra-cellular signaling and molecular mechanisms to differentially regulate specific sets of genes in the total genome, or the epigenetic control of cell division, differentiation, apoptosis or senescence; (f) a selective means to remove or kill specific cells in a multi-cellular organism without killing the organism (apoptosis or autophagy); and (g) senescence of both the cells and organism. These examples in Figure 1 also illustrate how oxidative stress, while allowing for DNA replication in these single cell organisms, blocks cytokinesis. The example of the anti-cancer drug cisplatin on *Escherichia coli* is shown because, when cancer cells are treated with this drug, genomic DNA replication continues, but the cells die because of the result of gap junctional intercellular communication [34]. In addition, in higher organisms, such as rodents and human beings, the liver contains hepatocytes of different ploidy levels. The explanation for this is that during the streaming of newly-formed hepatocytes, the oxygen levels change, leading to the blockage of cytokinesis but not DNA synthesis. The evolutionary significance of this change in oxidative stress in the liver, a detoxifying organ, was as an adaptive strategy to form a liver that was a better detoxifier. There were at least two ways by which a liver had a better means to detoxify toxins/toxicants. One was (a) to mutate the existing detoxifying genes to allow higher enzyme activity, or (b) to have more of the normal genes in a cell. When the hepatocyte can replicate its genomic DNA after mitogenic stimulation but does not have the ability to go through cytokinesis, this allows the cell to have extra copies of these genes, so it is more competent at detoxification.

The importance of the rise of oxygen in the environment is related to another driving force for organisms to survive. As the anaerobic single cell organisms found this new environment toxic, through a long series of evolutionary changes, of which the details are yet to be worked out, new biological systems, including the appearance of the mitochondria, which can metabolic glucose via oxidative phosphorylation to produce ATP much more efficiently than via glycolysis, and their symbiotic fusion with any early unknown cell, led to a cell that had many new genotypic/phenotypic characteristics, which led to multi-cellularity [38,39].

Clearly, this was not a one-time appearance of some unique gene, but, more likely, the slow accumulation of genes and phenotypes that allowed cells (a) to attach to each other; (b) to acquire growth control; (c) to differentiate into multiple functioning cells, such as the heart, blood, muscles, eyes, brain, kidneys, etc.); (d) to selectively die during development (apoptose); (e) to divide either by symmetrical cell division or by asymmetrical cell division or “stem cells”; (f) to form an oxygen-deficient micro-environment or a “niche” of the stem cells [40]; and (h) to be able to senesce [41]. With the new environmental appearance of oxygen, a family of molecules, such as the collagen family, now was available since this molecule needed oxygen to be synthesized [42]. With cells able to form collagen, they could now stick together. Of course, no one gene of gene function could satisfy meeting all these very different functional phenotypes. However, this then created another evolutionary driving force, since forming a clump or society of cells created conditions of unequalled ability of some cells to have equal access to nutrients or a means to eliminate metabolic toxic molecules. There also had to be some means of “growth control” or else this society of cells might be viewed as a “tumor”.

In the spirit of trying to provide at least one new function that, of course, leaps over the absence of hard scientific facts, a new means to acquire the ability to communicate in a new manner, namely, directly from one cell to its contiguous neighbor, had to meet the requirement of a society of cells to be adaptive to changing external environmental signals. The single cell organism had the ability to communicate with other members of its species via primitive hormone-like signaling or via “quorum sensing” [31] or “extra-cellular” signaling molecules. They also could regulate their gene activity when they encountered physical/chemical environmental factors through various “intra-cellular” signaling pathways. However, in this new colony of bound cells, a new method of signaling had to be formed. It is now speculated that the connexin gene family arose to meet that new need. In very early vertebrate multi-cellular metazoans, such as the sponge, this gap junction gene appeared [43]. Other genes and cellular structures that seemed to perform similar functions are the innexins and pannexins [44,45].

Today, in vertebrate metazoans, 20 connexin genes exist [46]. The connexin gene codes for a protein, the connexin, that can self-organize in the cell into a connexon hemi-channel unit consisting of six connexin proteins that can be transported to the cell membrane, where they can fuse with a neighboring connexon hemi-channel to for a gap junction channel, through which ions and small molecular weight regulator molecules can freely pass without the need to go through the membrane into intercellular space and then into the membrane of the neighboring cell [47,48]. Additionally, they can exist as “hemi-channels” [49].

Conceptually, one has to view these gap junction channels as not being formed as stainless steel or concrete channels. Rather, they were evolutionarily selected to be sensitive to physiological changes within the cell to regulate the transcriptional expression of this gene family [50], such that not all 20 genes are expressed in any single cell, but are expressed in a specific differentiated cell type [51], translationally regulated [52,53] and, most importantly, for quick responses to physiological or environmental changes, they could be post-translationally regulated [54,55]. By specific expression of one or a few connexin genes, the signals, which could be transferred through these channels, allowed for the differential expression of genes in those coupled cells, so as to allow unique differentiation of cells within the vertebrate metazoan.

At this point, one must now confront the issue of the role of gap junctions in embryonic/fetal development [56]. The single fertilized egg or zygote is, by definition, the toti-potent stem cell. It is in its most “undifferentiated” state. Early work showed that, as this fertilized egg starts to divide to form the blastocyst at the compaction stage, the gap junction starts to appear [57]. As Markert beautifully described, during the development of the embryo-fetus, each cell division creates new micro-environmental changes for these new cells [58]. These changes, in turn, alter gene expressions and cause the new differentiation of cell types, etc. Therefore, in this new embryo, different cell types (heart cells, blood cells, muscle cells, nerve cells, etc.) communicate with like-type cells via gap junctions to facilitate electrotonic or metabolic synchronization, while, at the same time, they can communicate between cell types via different extra-cellular signals to cells with various expressed receptors which, in turn, trigger specific intra-cellular signaling pathways. 

It is as though one could be watching individual members of the United Nations talking to other members of their delegation but, at certain times, listening to the General Assembly Director talking to the various delegations with one extracellular message. This complex method of integrating extra-, intra- and gap junctional inter-cellular communication is the means for homeostatic control of cell proliferation, cell differentiation, apoptosis and adaptive responses to occur [39]. It should be obvious at this point that any disruption in the multi-cellular organism, especially during development, that involves either lethal or abnormal viable development might occur if any of these three signal components are blocked.

To give but one example of how the differential expression of the total genome of a multi-cellular organism is regulated by unknown complex intercellular communication mechanisms, after the fertilization of the butterfly egg, the larva starts to differentiate cells to form mandibles for the larva to chew its food. The larva then forms a pupa where those larva tissues/structures/functions are removed by apoptosis to allow new structures to form, so when the butterfly emerges from the cocoon, it has wings to fly and a proboscis to obtain pollen from flowers. Similar, but obviously different inter-cellular mechanisms occur between a wide range of species, from fish to human beings, during development [59].

The development of knock-out/knock-in mice has shown that certain connexin genes are vital for life or can lead to altered development/function of the surviving organism [60]. Specific inherited mutations of specific connexin genes are also associated with various abnormal syndromes [61,62]. As will be demonstrated later, also, both endogenous and exogenous chemicals can modulate connexin gene expression or gap junction functions [63].

## 3. The Mechanisms of Carcinogenesis and the Role of Gap Junction Genes

With the exception of teratomas [64], most, if not all, cancers require a number of very distinct mechanistic steps for a normal cell to become an invasive and metastazing cell (i.e., the Hallmarks of Cancer [65,66]). A multi-stage, multi-mechanism concept of carcinogenesis was established by early studies [67,68]. The distinct phases of “initiation”, “promotion” and “progression” were operationally defined [6]. The “initiation” phase describes that when an animal is exposed to a “carcinogen”, a single cell in the exposed tissue is irreversibly converted to one that has the subsequent potential to become a cancer cell. It is assumed that the underlying molecular mechanism leading to that single cell having the potential of becoming cancerous is the induction of DNA damage and the formation of a mutation (“error of DNA replication”) [69,70,71,72,73]. On the other hand, a mutation can also be produced without DNA damage but as a result of an “error in DNA replication” of that single cell [74]. The functional result of a particular mutation in that single initiated cell is assumed to convert a “*mortal*” cell to become “*immortalized*”. Therefore, that single initiated “immortalized” cell could remain in the tissue for as long as it might take to accrue all the required genotypic/phenotypic changes required to finally convert it to an invasive and metastatic cancer cell (Hallmarks of Cancer [65,66]).

However, what was shown in those early animal studies is that the initiated cell has to be exposed to agents or conditions (wound healing, growth, tissue death) for some mechanism(s) to occur that selectively amplifies the “initiated” cell to form a benign lesion, such as a papilloma, enzyme-altered focus, nodule of the breast, or polyp in the colon. When it was shown that these “promoting” agents or conditions act via “epigenetic” mechanisms to (a) stimulate the mitogenesis of the initiated cells and to inhibit apoptosis of those initiated cells [75,76], it was hypothesized that this represents that the inhibition of cell–cell communication between the initiated cell and its contiguous and surrounding neighbors has occurred. Initially, one form of that “intercellular communication” was shown to be via “gap junctional intercellular communication” (GJIC) [26]. Since this “initiated cell” seemed to be “contact-inhibited” until exposed to agents/conditions that could inhibit contact inhibition, it has been postulated these “initiated” cells have functional gap junctions. Yet, as will be shown later, another form of cell–cell communication could also effectively inhibit a different kind of “initiated” cell [77].

Without dwelling on the “progression“ phase of carcinogenesis, where a clone of promoted initiated cells will accrue the final steps to convert the benign initiated cell to meet all the required “Hallmarks of Cancer” to invade tissue and to metastasize, it might seem appropriate to explain the promotion mechanism in more depth. Promoters can be characterized as either “endogenous” (e.g., hormones, growth factors, cytokines) or “exogenous” chemicals (pesticides, food additives, pharmaceutics, toxicants, food supplements), exogenous microbials (parasitic-, bacterial-toxins), or solid particles (e.g., asbestos, nanoparticles, etc.) [27]. Moreover, these promoting conditions are affected by strain and species [78], gender [79], developmental stage of exposure [80], threshold levels and concentrations [81,82,83,84], the regularity and sustained nature of exposures [84], and the absence of “anti-promoters” or “antioxidants” [85,86].

In terms of using the two major characteristics of “initiation” and “promotion” for purposes of cancer prevention or cancer therapy, it has to be pointed out that, while one can reduce the “initiating” event, such as to reduce one’s exposure to UV light from the sun, which is an efficient point mutagen or initiator, one cannot reduce “initiation” to zero risk because initiation can occur via errors in the DNA replication of a dividing cell. Therefore, since the promotion process requires exposures to long, sustained, uninterrupted and threshold levels of some promoting agent on tissues already exposed to an initiator, in the absence of anti-promoters, the most practical intervention strategies to prevent cancer occur during the “promotion” process.

Since all humans have “initiated” cells in all our organs, with more occurring as we age, sustained, chronic exposures to anti-promoters might seem to be a reasonable strategy. We probably get exposed to some endogenous (hormones, growth factors, cytokines, etc.) and exogenous (promoting food ingredients, toxins, pesticides, drugs, cigarette smoke) agents, some at below threshold levels; therefore, the promotion of any specific initiated cell might not occur. In addition, because each promoter type induces specific intracellular signaling, exposure to a generic “anti-promoter” or “antioxidant” might not counteract the signaling by that specific promoter [87]. As with the various oncogenes, such as *H-ras, Src, Neu*, their encoded mitogenic signing pathways are very different. Therefore, no single anti-oncogene inhibitor will work on all oncogenes. An anti-tumor promoter for phorbol ester induction of protein kinase C will not work to block promotion by phenobarbital.

The strategy to inhibit the promotion process by some anti-promoter might, at too high a concentration, actually end up acting as a promoter. This has been shown with retinoids [88,89]. In addition, it becomes even more complex as it has been shown that, even exposure to a known tumor promoter at its mitogenic threshold level will actually act as an anti-promoter when exceeding that threshold level. Estrogen, which seems to act as a breast tumor promoter by activating the estrogen receptor-signaling, can act as an anti-breast tumor promoter by activating non-receptor signaling that interferes with estrogen-dependent signaling [90,91]. In brief, chemicals, under different physiological cell conditions, can act as either oxidants or antioxidants [92]. 

Therefore, if one examines what is happening in the promoter-induced tissue lesion, one observes a heterogeneous array of genotypic/phenotypic cell types. As that lesion increases in size/volume, the micro-environment changes, which induces different gene expressions, causing the phenotypic differences seen in the different initiated clones. This, in turn, affects how they signal to each other and respond to whole body signals. 

This now sets the stage for asking the question: “What was the first normal cell that got ‘initiated’ which ultimately led to this heterogeneous collection of pre-cancerous cells, from which will emerge the ‘cancer stem cell’?”

## 4. What Normal Cell Can Be “Initiated?”

Within the historic context of the two major opposing hypotheses of carcinogenesis, namely, the “stem cell” [9,10,11,12,13,14] or the “de-differentiation or reprogramming” [15] hypotheses, and within the current hypothesis that the sustained ability of a cancer to grow indefinitely, the “cancer stem cell” hypothesis [93,94] now seems to be the target for both new strategies for prevention and treatment [95,96].

Conceptually, in developmental biology and embryology, the existence of stem cells was known. Yet, at that time, no one had isolated and molecularly—or biochemically—characterized any stem cell. Early molecular characterization had shown that the gene, *Oct4* [97], seemed to be the critical gene that needed to be expressed for a stem cell to be in the “undifferentiated” state. Fast forward, when a limited set of embryonic genes were transferred into a primary population of fibroblasts, a few clones of embryonic-like or “induced pluripotent” stem cells were isolated [18]. The interpretation of this experiment, which led to the Nobel Prize being received by Dr. S. Yamanaka [98], was that this mixture of genes was necessary to “reprogram” some “mortal” or differentiated fibroblast to become “immortalized”. The functional or operational definition of these “iPS” cells was that they must form teratomas when injected back into the syngeneic adult animal. 

Extrapolating these reproducible experiments to the cancer field, one must assume the multi-stage, multi-mechanism process occurs via this same presumed interpretation of “re-programming” or “de-differentiation”. Therefore, the “initiation” event must occur in this manner. However, some critical re-interpretations of the original “iPS” experiments and some new experimental findings have to be considered in another interpretation of the “initiation” and promotion” process in vivo.

The first argument is that if “initiation” was the result of a “re-programming” by some mutation that can convert a differentiated cell in any organ to become “immortalized” or become an “iPS”-like cell, then one should see mostly “teratomas” in adult animals after initiation, instead of carcinomas or sarcomas. That is not the case [99,100,101,102,103,104,105].

Second, while embryonic stem cells were isolated and partially characterized at the time of the “iPS” experiments [16,17], so, too, was the isolation of various organ-specific adult stem cells [20,21,22,23]. While there exist claims that normal stem cells do express connexins or have functional gap junctional intercellular communication [106,107,108,109,110,111], it seems that, because of real differences in the manner by which these experiments were done, compared to those where the normal adult stem cells were done, or because of the types of “cancer stem cells” studied (i.e., those derived directly from the stem cells or those obtained from tumors which might have had viral or endogenous oncogenes expressed. For example, in the paper by Hsiao et al. [107], the very early passage rat liver WB-F344 oval cells were a mixture containing few original oval cells, which were *Oct4A*-positive but Cx-negative, and with many *Oct4*-negative, but *Cx43*-positive functional GJIC cells. Late passage cells have very few *Oct4A* and Cx negative cells. In vitro, this cell line ultimately senesces. The papers claiming that embryonic stem (ES) cells have expressed connexins or functional GJIC have additional problems due to the nature of the in vitro culturing conditions. However, since these cells do grow on confluent feeder layer cells that have functional GJIC, then one might expect these ES GJIC-positive cells to be “contact inhibited”, to apoptose or to differentiate. These ES clones could, in time, start to differentiate and actually be a mixture of cell types. In the intact embryo, it has been shown [56,57] that the connexins and functional GJIC appear only after the “compaction stage”. If these GJIC-positive ES cells are growing on the feeder layer, there has to be an explanation. Until further experiments are done that show there is actual GJIC between these two different cells, one must await a resolution to this dilemma. Finally, in non-adherent cells, such as blood cells, lymphatic cells, neutrophils, etc., having functional gap junctions is not possible in those stages when they must be “free floating” and migrating. While leukocytes do not express Cx’s, when stimulated with endotoxins, they can express the connexins and form leukocyte–endothelial cell and leukocyte–leukocyte contacts [112]. During these short periods of time, these cells can transiently express their connexin genes.

It does seem, from studies with both human adult stem cells and the cancer stem cells used in this “Concept Paper” [21,113,114,115,116], that the normal adult stem cells do not express connexin genes or have functional gap junctions and that “cancer stem cells” can be of two types: those that do not express connexins and those that do express connexins but have no functional gap junctions. It must be noted that normal stem cell types need to be isolated and grown on confluent “feeder layer” cells. If those stem cells had functional GJIC, they would be contact-inhibited by the feeder layer cells. Cancer cells, on the other hand, that do not have functional gap junctions can grow on these feeder layer cells, which do have functional gap junctions (see Figures 4 and 5 in [20]). 

With these adult human breast stem cells [21], it was shown that those that express the estrogen receptor do not express connexins or have functional gap junctions but do express *Oct4A* (see Figure 2, [13]). Moreover, they did not form teratomas when put in immune-deficient mice, but when placed on matrigel, they could form breast tissue-like organ structures [116]. Also, when suspended in non-adherent culture conditions or in soft agar, these normal breast stem cells could form mammospheres that grew to a certain size before stopping growth [117,118].

More importantly, if these cells were exposed to the *Src* oncogene, they seemed to have their ability to differentiate in an inhibited manner, i.e., they were “initiated” [21]. Since the definition of “initiation” is the induction of “immortality”, which is the inhibition of differentiation or “mortality”, then these *Src*-treated breast stem cells were “initiated”. More importantly, a normal stem cell is naturally “immortal” until it is induced to differentiate or to become “mortal”. Therefore, what happened in this case was that the insertion of the *Src* gene into the genome of the normal undifferentiated breast adult stem cell blocked the “mortalization” of a normal “immortal” cell. This radical re-interpretation of how “immortalizing” viruses, such as Src, HPV, etc., is not to immortalize a “mortal” differentiated cell to become “immortal’, but to block the “mortalization” of a normal “immortal” cell. Reprogramming did not occur in this case. The *Src* gene did not turn on the *Oct4A* gene, but kept the gene expressed [13].

In this series of experiments where these *Src*-transformed breast stem cells could now be exposed to radiation (a known human breast carcinogen), which led to a few of these initiated stem cells that formed colonies that were still expressing the *Oct4A* gene and were now weakly tumorigenic. After these cells had been genetically transformed with the *Neu* oncogene, they became highly tumorigenic and continued to express the *Oct4A* gene. Significantly, the normal differentiated breast epithelial cells, which were derived from the normal adult breast stem cells, no longer expressed the *Oct4A* gene and exposure to the *Src* gene did not give rise to any “immortalized” cells or any cells with their *Oct4A* gene expressed. In other words, the differentiated breast stem cells were not “re-programmed”. This only reinforced classical studies of the failure to “immortalize” human cells and is supported by the recent demonstration by Dezawa on the human multilineage differentiating stress enduring cells (Muse cells) cell experiments [118].

The demonstration that organ-specific adult stem cells appear to be the “target cells” for initiating the carcinogenic process [13] suggests that they are the origin of the “cancer stem cells”. If this hypothesis is correct, then after the initiation of an organ-specific adult stem cell and its subsequent promotion by any tumor promoting mechanism, a clone of benign pre-malignant cells should appear. Only after continued promotion and the lack of apoptosis, more genetic and epigenetic alterations might be expected to occur to give rise to a premalignant tumor whose cells are heterogeneous [119,120]. Out of that heterogeneous population, including the original initiated stem cell, a cell that has accrued all the “Hallmarks of Cancer” might be expected to appear and would escape the confines of that tumor and invade and metastasize. 

From the early work on rat liver tumorigenesis, the observation that the liver diploid oval cell, which gives rise to both the hepatocyte and bile epithelial cell lineages, seems to be the target cell that is initiated. Since the hepatocytes derived from the oval cell go through a series of endoreduplications that lead to a streaming population of polyploidy cells (immature hepatocytes are diploid and increasing mature hepatocytes are tetraploid and octoploid) [120,121,122,123,124,125]. This observation seems to mimic the phenomenon in *E. coli* shown in Figure 1. Yet, the newly formed “enzyme altered foci” or the promoted initiated oval cells are diploid [125]. This strongly suggests that the tetraploid and octaploid hepatocytes are not the target cells for initiation, because the “initiation” event in the “de-differentiation” or “reprogramming” hypothesis would have to bring about a diploidization of a polyploidy cell. This is highly unlikely and is a phenomenon that has never been observed.

In the cause of human liver “oval” cells which have been isolated and characterized, they express the *Oct4A* gene but do not express the *connexin 43*, *26* or *32* genes or have functional gap junctions [126]. On the other hand, the normal non-tumorigenic and diploid rat liver cells have been shown to express the *Cx43* gene and have functional gap junctions. These cells can be neoplastically-transformed, especially with various oncogenes [29]. This now sets the stage for the main point of this “concept paper”, namely, that there are two types of cells that can become gap junction deficient cancer cells. This is a major point that was not noticed by Loewenstein and Kanno [5], nor has it been a major issue in the general cancer field of cancer prevention and treatment.

## 5. How Two Types of Gap Junction Intercellular Communication—Deficient Cells Require New Strategies for Prevention and Treatment of Cancers

From just a simple strategy, assuming Loewenstein and Kanno’s idea was correct, namely, that all cancer cells that do not have “contact inhibition” do not terminally differentiate or have normal apoptosis and are “immortalized” so that they cannot senesce, then the strategy should be to get these cancer cells to restore their ability to communicate via gap junctions. However, it is assumed that (a) some “cancer stem cells” are derived from a normal organ-specific adult stem cell that does not express its connexin genes or have functional GJIC, as in the case of the human breast cancer demonstration [115]. Yet, as shown by the rat liver examples, the oval cells that do express *connexin 43* are target cells for liver carcinogenesis, in that the initiated and promoted clones are both partially differentiated into hepatocyte-like cells (Potter’s “Ontogeny as partially blocked ontogeny” [11]). Consequently, in the total universe of all organ system tumors, one should find “cancer stem cells” within a heterogeneous population of “cancer non-stem cells” that are partially differentiated. 

To assume that one needs only to target prevention and therapeutic strategies on “cancer stem cells”, which is, in general, a great place to start [127,128,129], is not sufficient. The point is that the target cell for the “cancer initiated cell” that led to the “cancer stem cell” might have been (a) a normal adult stem cell that never expressed its connexin gene, because it had expressed its *Oct4A* gene, or (b) one that was a very early immature differentiated stem cell that transcriptionally shut down its *Oct4A* and transcriptionally turned on a connexin gene, but also had an active oncogene transcribed that rendered the gap junctions inoperative [28,29]. Clearly, at this time, there is no direct evidence that the *Oct4A* transcription factor directly regulates or represses the transcription of the connexin gene family. Yet, it can only be inferred that these two important gene families are strongly related. *Oct4A* is needed for “stemness” and for being able to remain “undifferentiated” or not being able to divide asymmetrically, whereas the transcription of the connexin gene and the formation of functional gap junctional communication seems to be required for asymmetrical cell division and differentiation. This assumption is logically derived from the experimental observations that cancer cells, which do not “contact inhibit” or have growth control, do not terminally differentiate, do not apoptose normally and seem to have been “immortalized” and do not have functional gap junctional intercellular communication. Future studies will need to determine the exact role that the Oct4A transcription factor has with the connexin gene expression. However, it is known that *Src* and HPV coded gene products can render p53 and RB gene products to be non-functional [130].

So here is the new insight for potential strategies for cancer prevention and treatment. If one assumes all “cancer stem cells”, in terms of their inability to perform functional gap junctions, are the same, then what one will discover is that the same anti-cancer drug will not work on both types [77]. The explanation is this: (a) If an organ-specific adult stem cell is “initiated” by some mutational event that prohibits the *Oct4A* gene from being transcriptionally repressed, then that cell will not be able to activate the genes needed for differentiation, such as the connexin genes, which are needed to form the gap junction channels These cancer cells would tend to be more “embryonic-like tumor cells”.

If that “initiation” event in the adult stem cell is the activation of an oncogene, these proteins have been shown to render the *p53* and RB gene proteins inactive, thereby blocking differentiation. On the other hand, if the organ-specific adult stem cell has repressed the *Oct4A* gene and started to transcribe the connexin genes, but at the same time also transcribes any oncogenic oncogene, such as the *H-ras* or *Src* gene, these oncogene proteins can render the coded connexin proteins to be non-functional by hyper-phosphorylation of the connexin protein [29].

Consequently, treating either of these non-communicating cancer cells with the same agent designed to restore gap junctional intercellular communication will fail to work on one class. In order to restore GJIC in the adult stem cell that has its *OCT4A* gene transcribed, one would need some agent that could transcriptionally repress the *OCT4A* gene. Possibly, drugs, such as SAHA [131], might work if they could be targeted to the “cancer stem cells” expressing *OCT4A*. On the other hand, for those “cancer stem cells” that have already suppressed the *OCT4A* gene and started to express the connexin proteins, one would need to find inhibitors to the specific oncogene protein, such as tyrosine kinase or to activate a protein phosphatase, in order to restore the gap junction protein to its functional state.

Therefore, while it is absolutely true that both genotypic and phenotypic heterogeneity with the tumor that was derived from a single cell organ-specific adult stem cell or its very early immature differentiated diploid derivative exist, the important point is that not that heterogeneity is the crux of the problem, it is the genotypes and phenotypes of the “cancer stem cells”. It would seem a practical strategy that the initial pathological examination of a tumor would involve determining whether the “cancer stem cells” of a tumor express *OCT4A* and no connexin protein or if they do not express *OCT4A* but do express some connexin protein. Since neither “cancer stem cell” has functional GJIC according to the Loewenstein hypothesis, the treatment strategy would be either (a) to repress the *OCT4A* gene to induce GJIC or in the other case (b) to reverse the posttranslational modification of the connexin protein, in order to restore normal GJIC. Finally, the use of normal human liver stem cells, grown in three dimensions by eliminating the use of animals or two dimensions through the use of either primary hepatocytes or various liver cancer cells, should be the next wave of screening for agents affecting the human liver [132].

## 6. Summary of the Roles of Stem Cells and Gap Junctional Intercellular Communication in Normal and Abnormal Growth Control and Differentiation

During the early evolution of multi-cellularity in vertebrate metazoans, the gap junction gene family appeared at the time when growth control, cellular differentiation and apoptosis cell functions appeared. The environmental role of oxygen played a major role in the co-selection of other genes that allowed (a) the formation of stem cells in low oxygen microenvironments to keep the stem cells in their “primitive state, i.e., their niches; (b) the genes that controlled the decision process of symmetrical cell division to maintain “stemness” of both daughter cells and asymmetrical cell division of these stem cells to allow one daughter to remain “stem-like” and the other to start to terminally differentiate; (c) the regulation of genes, needed for the differential gene expression of the genome, including only one or few connexin genes, to express only those genes required for specific cell differentiation; (d) the induction of selective cell death or apoptosis; and (e) the triggering of senescence. 

From the single “toti-potent” fertilized egg until the fully-formed adult organism, in all the trillions of cells in different organ and tissue types, three forms of cellular communication exist, namely, extra-, intra- and gap junctional inter-cellular communication. Homeostatic control of cell proliferation, differentiation and apoptosis of the three cell types—stem, life-span limited progenitor cells and the terminally differentiated cells—must be maintained, especially during early embryonic and fetal development. In the case of carcinogenesis, all it takes is for one organ-specific adult stem cell to lose its ability to have functional gap junctional intercellular communication, either because it could not transcriptionally express its connexin genes or because it posttranslationally modified its connexin proteins for functional gap junctions for that cell to become “initiated”. In either type of “initiated” cell, depending on exposure to agents/conditions that inhibit extra-cellular communication or gap junctional inter-cellular communication, it can become an invasive and metastatic cancer cell. As those “promoted and initiated pre-malignant” cells grow, they alter their internal microenvironments, so that some of these initiated cells phenotypically change due to partial differentiation to produce the tumor heterogeneity and some additional genotypic changes in these initiated cells occur to accrue the ultimate “hallmarks” of a “cancer stem cell”. By being able to distinguish between the “cancer stem cell”, which never expresses its connexin genes or represses its *OCT4A* gene and those “cancer cells” that do repress their *OCT4A* gene and transcriptionally express their connexin gene(s), one might be able to restore “normal” GJIC in those two types of “cancer stem cell” as a strategy for cancer treatments.

## Figures and Tables

**Figure 1 cancers-11-00087-f001:**
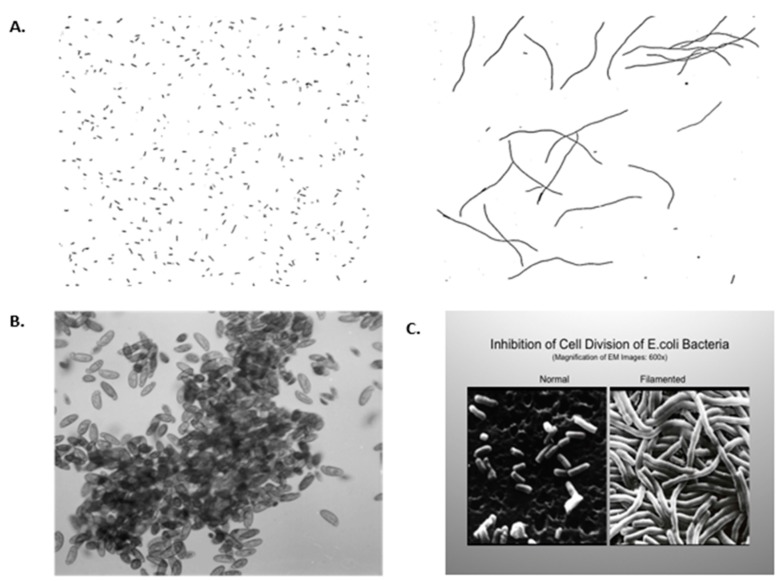
(**A**) Filamentation of aerobically grown Hpx-mutants of *Escherichia coli* (*E. coli**)* cells. Cells were grown in Luria broth anaerobically (**A**) or aerobically (**B**). Magnification: ×400. [35] Permission granted by Proc Natl. Acad. Sci., (PNAS). (**B**) J.M. Saul, Lethaia, 2008: Clumping of anaerobic cilates in oxygenated water [36]. Permission granted by Les 3 Colonnes, Paris. (**C**) *E. coli*, grown in traditional growth medium, showing normal morphology. Magnification: ×600. When *E. coli* were grown in the same medium but with a submerged platinum electrode, the *E. coli* had their DNA replicate, but they did not septate. This observation led to Dr. Barnett Rosenberg’s discovery of the anti-cancer drug cisplatin [37]. Permission granted by: Paul Rosenberg of the Board of Barros Foundation. Permission granted by Springer Nature, N.Y.

**Figure 2 cancers-11-00087-f002:**
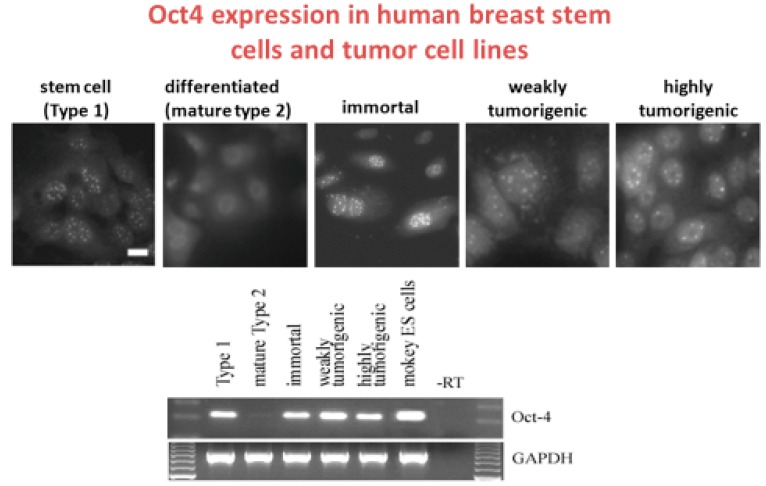
The composite of these two figures illustrates that the clonally- derived normal human adult breast stem cells express Oct4A (Type 1), via immune-histochemical use of fluorescent antibodies to Oct4A in the normal breast stem cells, but not in the differentiated breast cells (Type 2), are still expressed in SV40 immortalized normal breast stem cells, which are not tumorigenic, are still expressed in irradiated and weakly tumorigenic breast stem cells and are expressed in the neu/ErB-2 highly transformed clone. The Type 2 cells were differentiated normal human breast stem cells. The reverse transcription polymerase chain reaction data are correlated with the immuno-histochemical data of Oct4A in this series of human breast adult stem cells. No expression of Oct4 was seen in Type II differentiated cells (lane 2) but Oct4A was still expressed in the immortalized, weakly and highly tumorigenic cells. Monkey ES cells were used as a positive control (lane 6). Lane 7 is a no template control. While the GJIC data were not shown here, the normal breast adult stem cells (Type 1), immortalized, weakly and highly tumorigenic cells have no functional GJIC, as measured by the fluorescent scrape loading/dye transfer technique, whereas the Type 2 cells had functional GJIC. Type 2 or differentiated Type 1 cells were derived from Type 1 or adult human breast stem cells after the Type 1 cells had been exposed to cholera toxin [114]. Scale bars: For Type 1 to highly tumorigenic cells was 20 μm. Permission granted by MDPI Publisher, Basel, Switzerland.

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
