# Peer review of "Cancer Prevention and Therapy of Two Types of Gap Junctional Intercellular Communication–Deficient “Cancer Stem Cell”"

_cancers, 2019, doi:10.3390/cancers11010087_

Reviewer 1 Report

The manuscript titled “Cancer Prevention & Therapy of Two Types of Gap Junctional Intercellular Communication–Deficient  “Cancer Stem Cells”” by JE Trosko, is a brilliant historical excursion on the development of studies on the carcinogenic processes and on the role that the cell-cell communication mediated by gap junction was attributed (or not) in that processes. In addition, the manuscript offers an clear and original view of the argument.

 The field of the mechanism(s) of carcinogenesis and gap junction cell-cell communications is faced from the wider point of view, from molecular to functional studies, pointing crucial relationships.

This reviewer has read critically the paper, trying to find some weaknesses, but none robust objections was possible. In this paper it is possible to found concepts that could really add information and interest to the field of relationship between the cancer processes and the role of this basic cell function. 

This reviewer recommends the publication of the manuscript  in the present form in this Special Issue of "Cancer". 

Author Response

 I very much appreciate the praise given to the purpose of my "concept" paper:  "brilliant historical excursion...".

I will, together with comments of the other two reviewer , go over, those English style problems.

Reviewer 2 Report

Please verify that the references cited support the statements that are made in the text.  For example, line 321, reference 116 does not, I believe, discuss Cx43, 26 and 32.

Is Figure 1 necessary?

A schematic diagram would be helpful for the reader.

Author Response

To reviewer #2: I have corrected the reference to which the comment was raised.

To answer the comment, "Is figure 1 necessary?" , my answer is absolutely! In this Concept Paper, I am trying to point out that the whole issue of cancer causation and treatment must be seen in the context of "evolution", and in modern terms, human diseases must be seen in the concept of "Evolutionary Medicine". . Finally, multicellular metazoans, such as the human being , dependents on the normal functioning of gap junctional intercellular communication to maintain homeostatic control of cell proliferation, differentiation and apoptosis. Any disruption of that fundamental biological function, all kinds of diseases can result. While that figure might seen abstract, my experience with the discovery of cisplatin showing that it dose not work as an anti-cancer drug by damaging genomic DNA, but works, epigenetically, to block cytokinesis and to involve gap junctional intercellular communication, as noted with recent reference, it also gives a clue to the evolutionary origin of normal hepatocyte polyploidization for the liver to be a better toxicological filter. 

I have expanded this explanation a bit in the revision.

As to adding a schematic figure, I am not sure I understand what is suggested and therefore, without more specific suggestion, I do not feel adding another figure is warranted.

Reviewer 3 Report

This “concept paper” discusses a very interesting topic of interest to the field.

However, major restructuring and adaptations, including  correction of factual errors and inclusion of more solid arguments not only for but against the hypothesis, needs to be included / addressed before this manuscript can be reconsidered to be suitable for publication.   

Overall both figures are virtually useless and rather a simple diagram illustrating the hypothesis is needed.The interest of the author sometimes goes way beyond the topic covered (or at least where the focus should be), leading to a poorly focused article that easily loses the interest of the reader.

Specifics:

Section2:  Most of the text is completely irrelevant to the actual hypothesis proposed, and could be reduced by 90% (e.g. lines 87-93 discussing oxygen/mitochondria etc etc..)

Figure 1 is totally irrelevant.

There are major factual errors. E,g.  

Line116…:

“It is now speculated that the connexin gene family arose to meet that new need. In very early multi-cellular organisms, such as the sponge this gap junction gene appeared.”

--Connexins DID NOT arise in the early multicellular organisms. Non-chordate multicellular organisms do not express connexins, but innexins, which are very different at the sequence level.

Line 118 “Today in metazoans, there exists 20 connexin genes”

--Again, not all metazoans express connexins! Only chordate animals do, and the number of genes differ between species, e.g. humans are thought to express 21 connexin genes in up-to-date databases (not counting pseudogenes),

Line144: this analogy is really strange.

Line 152:  “To give but one example, after the fertilization of the butterfly egg…..”

--Again, an example of an organism not expressing any connexin!

Section3: could be cut in half or more, it is too long and does not convey the simple precise take-home message that is needed to introduce the main concept.

Section4: could be cut in half or more, it is too long and does not convey the simple precise take-home message that is needed to introduce the main concept (which is introduced almost at the end of the paper!)

Figure 2 is really not relevant to the hypothesis. Better save this space for a figure delineating the hypothesis.

notes on some of the key claims:

e,g, Line 261: “...not express connexins or have functional gap junctions, but did express Oct4A [105,106]”

---This key claim needs to be significantly substantiated:

More specific references for this claim needed. Ref 105 (which is identical to reference 21  ???): does not extensively investigate the stem cell population. The authors show two morphological subpopulation, one of which is GJIC deficient and can form soft agar colonies upon SV40 transformation. This is very weak evidence for “cancer stem cells” being GJIC-deficient considering cancer stem cells typically constitute a minor fraction of the cancer cell population. REf 106 does not test or show that the stem cells are GJIC-deficient  (the entire HeLa cell population is GJIC-deficient, and the “oct4-specific” subpopulations of MCF-7 cells was not compared to the “non-stem cell” population.

Secondly; Several recent reports in well-renowned journals, suggest that tissue stem cells and cancer stem cells indeed specifically express connexins and even require them to stay in a CSC state.

Just a few recent examples:

normal tissue stem cells:

Stem Cells Dev. 2013 Nov 1;22(21):2906-14. doi: 10.1089/scd.2013.0090. Isolation of pluripotent neural crest-derived stem cells from adult human tissues by connexin-43 enrichment.

Kaohsiung J Med Sci. 2014 Feb;30(2):57-67. doi: 10.1016/j.kjms.2013.10.002. Inorganic arsenic trioxide induces gap junction loss in association with the downregulation of connexin43 and E-cadherin in rat hepatic "stem-like" cells.

CSC:

Cx46 specifically expressed and needed for glioma cancer stem cells:

Cell Rep. 2015 May 19;11(7):1031-42. doi: 10.1016/j.celrep.2015.04.021.
Differential connexin function enhances self-renewal in glioblastoma.

Cx25 in leukemia stem cells:

Oncotarget. 2015 Oct 13;6(31):31508-21. doi: 10.18632/oncotarget.5226.
Cx25 contributes to leukemia cell communication and chemosensitivity.

Cx43 in glioblastoma stem cells:
Oncotarget. 2016 Dec 27;7(52):86406-86419. doi: 10.18632/oncotarget.13415.
Patient-derived glioblastoma stem cells respond differentially to targeted therapies.

Cx43 in lung CSC...connexin 43, a constituent of GJs, in A549 CSCs.:

Anticancer Res. 2018 Sep;38(9):5093-5099. doi: 10.21873/anticanres.12829.
AS602801, an Anti-Cancer Stem Cell Drug Candidate, Suppresses Gap-junction Communication Between Lung Cancer Stem Cells and Astrocytes.

These are just some examples of litterature that seems ignored

Line 325  “the main point introduced“….⅔ into the paper, way to late…..

Line 329: “From just a simple strategy, assuming Loewenstein and Kanno’s idea was correct, namely, all cancer cells, which do not have “contact inhibition”, do not terminally differentiate, or have normal apoptosis, and are “immortalized” so that they cannot senesce, then the strategy should be to get these cancer cells to restore their ability to communicate via gap junctions.”

--This is not exactly the “idea” of Loewenstein.

Line 332:

“we believe that (a) some “cancer stem cells” are derived from a normal organ-specific adult stem cell, which does not express their connexin genes or have functional GJIC, as in the case of the human breast cancer demonstration [106].”

--Where is the evidence for this statement? Reference 106 fails to support this evidence. See also select references above that disagrees with this.

Line 341_

“The point being is that the target cell for the “cancer initiated cell” that led to the “cancer stem cell” might have been a normal adult stem cell that never expressed its connexin gene because it had expressed its Oct4A gene”

--Where is the evidence that Oct4 suppresses connexins? There is now “massive” evidence that ES cells and IPS cells that express huge amounts of Oct4, and at the same time express huge amounts of Cx43 (and other connexins) and are well coupled (as simple pubmed search reveals many papers)

Line 348: “If one assumes all “cancer stem cells”, in terms of their inability to perform GJ

….” Again, see references above, but one does not assume in science. There are arguments both for and against (and it seems mainly against), but this is not covered in this paper,

In general; the key point of the paper is not substantiated sufficiently: e.g:

Failure to show evidence that cancer stem cells (or adult stem cells for that matter) have loss of GJIC. (no robust papers are presented to support this, whereas several publications arguing against this theory are ignored)

Failure to show Oct4 regulates connexin expression (negatively).

Failure to recognise many tumour cells express high levels of Connexins and functional GJIC.  (in particularly the CSC subpopulation in some cases

There is overwhelming evidence many tumours re-form GJIC at the metastatic site and that this promotes tumour growth (recent clinical trials are now aiming to block GJIC in these tumours!)

This is totally inconsistent to this hypothesis. Yet it is not covered at all.

Overall,

This paper raises an interesting topic but the hypothesis is poorly presented due to two main reasons; a) being quite unfocused; b) being severely biased towards the hypothesis, ignoring the overall evidence in the case which points to the fact that this hypothesis is likely only relevant to some specific cancer cases (if any). The lack of reflection creates a conflict in which this commentary is rather misleading than truly informative.

Author Response

First, I wish to acknowledge the knowledgeable review by this expert in gap junctions. It is a review with "constructive criticisms" , that can, ultimately, improve a manuscript. I have taken seriously many of the comments/suggestions and have modified the manuscript to account for those I felt were justified. On the other hand, there were a few to which I fundamentally disagreed. I believe these disagreements come, primarily, from my 50 years of doing mechanistic cancer research, with stem cells and gap junctions, as a means to understand the origin, prevention and potential treatment of cancers. In addition, this paper was a "Concept" paper, addressing, for the very first time, a hypothesis that past & current failures in cancer treatment is due to not recognizing there are two very different kinds of "cancer stem cells". NO ONE in the CANCER FIELD HAS ADDRESSED THIS PROBLEM!!!!.

Point by Point , here are my responses.

1. Thanks to this reviewer # 3. for pointing out this paper does "...discuss a very interesting topic of interest to the field." My reason for doing it was because no one in this current field, including those that know "cancer stem cells" must be the target for future cancer therapy, has any insight to how to approach the problem. This is due, in large part, because all those in the field of cancer therapy do not consider any role that gap junctions play. One only need  to read the papers of the Biggies of the cancer field (Varmus; Bishop, Weinstein, etc.) to notice they never make reference to any of the gap junction giants.

2. I will try to address these legitimate "factual " concerns and supporting references.

3. Reviewer states the figures are useless and I should replace it with another diagram illustrating the hypothesis. I totally disagree because In this "Concept" paper, I am trying to point out that evolution, including multi-cellularity in vertebrate metazoans, helped to introduce genes that contributed to regulation of the homeostatic control of cell proliferation, differentiation, and apoptosis. That included the evolution of the connexin gene family. Moreover, today, using the concept of "evolutionary medicine", the idea that dysfunction of GJIC could contribute to many diseases should be a fundamental objective.

Figure 1. demonstrates how the evolutionary transition of no to low oxygen levels to that of normoxia led to the biological production of collagen for cells to "glue" to each other, and for stem cells, stem cell niches and for  the switch from glucose metabolism via glycolysis to oxidative phosphorylation to occur.  Moreover, my years of cancer experience with the discovery of the anti-cancer drug, cisplatin,  showed that it only affected some cancer cells, but not all, within a tumor. More recently, it has been demonstrated that  gap junctions are involved with the future treatments of the heterogenous nature of all cancers ( "cancer stem cells" and "cancer non-stem cells"), to which I added a recent reference. Moreover, the reason for the demonstration of how cisplatin worked to induce "spighetti bacteria" by inducing the blockage of cytokinesis after DNA replication was to  illustrate the role of oxygen (oxidative stress) in the transition of single cell organisms to multicellularity, the role of oxygen in stem cell differentiation and the role of oxygen in the expression of connexin genes. This phenomenon in E. coli might be argued as a primitive form of "differentiation" or "epigenetic regulation of the phenotype in a single cell organism. I will add more discussion to clarify the use of the Figure 1. It will also be linked to a discussion later in the chapter. As an aside, the current thinking about the mechanism of cisplatin toxicity on cancer cells is that it damages genomic DNA of the cancer cell. This is not correct!!!. To have this figure in this Concept paper on two types of "cancer stem cells" should be a major component.

I will modify discussions on both Figures to justify their use.

4. Reviewer states"Section 2:  Most of the text is completely irrelevant to the actual hypothesis proposed, and could be reduced by 90% (e.g. lines 87-93 discussing oxygen/mitochondria etc etc..). I again disagree because in this Concept paper, this is critical to both the cancer field and gap junction field. Oxygen is key to stem cells, gap junctions, and glucose metabolism ("Warburg hypothesis") in normal stem cells and cancer stem cells. Oct4A is redox- sensitive. Oxygen levels regulate stem cell ability to divide symmetrically or asymmetrically.  I will modify  this section to see if it can't be improved. However, I cannot not eliminate it for my story.

5. Line 116: Factor errors are corrected with the suggested corrections.

6. Line 118 . Reviewer is correct and modification is made in the revised manuscript.

7. Line 144: Reviewer comments that analogy is "strange". I will expand on this point.

8. line 152; The use of the development of the butterfly was not to point out the role of connexins but that multi-cellular development had to utilize new genes/functions to regulate the total genomic information, epigenetically. The recent book , Your Inner Fish, by Neil Shubin illustrates the same concept. I added that reference.

9.Section 3.  Garth Nicholson

10. Sect 4. Reviewer suggests cutting it dramatically. Again, I disagree.

11. Figure two is irrelevant. I totally disagree with this reviewer. I will not cut this figure because it is the MOST relevant evidence to my hypothesis.

12. Line 261: Have made some adjustments to these concerns.

13. "Secondly; Several recent reports in well-renowned journals, suggest that tissue stem cells and cancer stem cells indeed specifically express connexins and even require them to stay in a CSC state."

Since I disagree with their interpretations, but will include these references with a few reasons for my diagreent with those references. My first point of disagreement is that those who claim the ES cell has expressed GJ's or functional GJIC ignore the fact that these cells only can be isolated and grow on confluent feeder layer cells ( that have functional GJIC. If these ES cells has functional GJs they woulf be "contact inhibited", apoptose or differentiate. If reviewer#3 had looked at our Cancer Research paper (1987), he/she will see that our human kidney stem cells do not have GJIC but the underlying  confluent human fibroblast cells do have functional GJIC. When normal GJIC functional human differentiated progenitor breast epithelial cells are placed on these confluent feeder layers, they do not grow, but human cancer cells, without functional GJIC, do grow.

Second, in the case of the rat liver "oval" WB-F344 cell reference, that group received those cells from my lab. We know that the very early passage of these  cells are heterogeneous mixture of  a few true liver stem cells ( CX-, Oct4A+). The later passage has very few of the Oct4A and Cx43- cells. These cultures eventually senesce because there are no stem cells left because of many passages in 20% oxygen. We have also isolated the true human kiver oval or stem cell. It expresses Oct4A, but does not express any connexins or have functional gap junctions [ Chang, C. C., Tsai, J. L., Kuo, K. K., Wang, K. H., Chiang, C. H., Kao, A. P., Tai, M. H., and Trosko, J. E. (2004). Expression of Oct-4, alpha fetoprotein and vimentin and lack of gap-junctional intercellular communication (GJIC) as common phenotypes for human adult liver stem cells and hepatoma cells. Proc. Am. Assoc. Cancer Res. 45, 642.]

Third, in the case of leukemia cells, one needs to recognize that there are cells in our body that are non-adherent, e.g., blood cells, lymphatic cells, neutrophils, etc. They need to be "free floating". During their immature phase, they must touch, transiently, their "nurse" cells , where they form GJIC , receive some differentiation signal, differentiate, disconnect and no longer need GJ's during their normal migration activities. My work with Garth Nicholson, an expert  on metastatic cells also showed these metastatic cells did not have functional GJ's. (  Nicolson, G.L., K.M. Dulski and J.E. Trosko: "Loss of intercellular junctional communication correlates with metastatic potential in mammary adeno­carcinoma cells.  Proc. Natl. Acad. Sci. 85:473‑476, 1988.  Nicholson, G.L., Lichtner, R.B., Trosko, J.E., Cytoskeletal and junctional heterogeneity in mammary tumor cells and their possible significance" In: tumor progression. Advances in Experimental Medicine and Biology, 233: 21-26, 1988).The "seed-soil" hypothesis now seems to indicate the metastatic cells go to many organs, can "contact inhibit" in some organs, re-express their GJ's sand do not grow.  However, in those organs where these metastatic cells do grow, the microenvironment suppresses the GJ's .

Clearly, if I need to explain all my disagreements with the interpretations of those suggested references ( and more that I know of), it would be another manuscript.

14. Line 325: I disagree.

15. Line 329. I modified this statement.

16. Line 332. Reviewer challenges this statement. I have re-written it.

17.Line 341; Secondly; Several recent reports in well-renowned journals, suggest that tissue stem cells and cancer stem cells indeed specifically express connexins and even require them to stay in a CSC state. I am aware of those references, and several more, but I disagree with their interpretations. None of them worked on single isolated stem cells. Second, the connexin genes in some cancer cells, which are suppressed in one state can be induced by invasion and metastatic connect. I have modified that manuscript (as above ).

18. Line 348. Again, I have modified the statements but I totally disagree with the interpretation of the results in their papers.

19. Reviewer summarized his/her conclusions that claim to contradict my hypothesis without pointing out the strength of all our papers showing the opposite of what is claimed in our peer-reviewed papers. This also includes our work with metastatis . See my Garth Nicholson references above

In science, no study is ever "complete". Many differences seen in the literature can be due to two differently executed experiments; two similar experiments but interpreted differently; One experiment is done correctly, the other is not, etc.

While this reviewer has made valuable suggestions and offered correct references and offered alternative explanations, I feel, I have accept those that I feel are correct and relevant, but I stand by my 50 years of experience in this field and I am free to offer my opinion or interpretation to the scientific community. The scientific community has to be the judge. However, my "concept" paper does explain the current reason our cancer therapy has not worked and I have offered a unique explanation that does not yet exist in the literature. The reviewer has not offered any explanation as to why my idea is wrong nor has he/she offered one that is better than any that exists.

Round  2

Reviewer 3 Report

I appreciate the lengthy answer from the author, and have carefully tried to understand the viewpoints of the author with an open mind. Nevertheless, there are several outstanding issues that I believe have not been addressed in a satisfactory manner (to me personally, not necessarily the editors).

I am very open for new perspectives and especially those that challenge the current view. However, I also believe that convincing evidence needs to be clearly demonstrated, together with  a balanced and open-minded view of litterature having an “alternative” view. In my view the current perspective is unbalanced in this respect.  

I will simply mention some key points that I believe could have improved the paper substantially. In the end however, it is up to the author and the editors in conjunction to weigh in on the importance of these mere suggestions.

I still believe a schematic figure is very important in order to convey the exact hypothesis in a clear manner. Personally I fail to understand the real value of the current figures. The author claims figure 1 is important to illustrate how “evolution introduced genes that contribute to proliferation etc.” and how this affect “differentiation” and “epigenetic regulation” etc.. I truthfully do not understand how some “morphological” pictures of bacteria (that do not express connexins let alone gap junctions, and are not even multicellular organism) have anything to do with this at all. The figure does not help me in understanding this paper whatsoever, and even though the author disagrees, I simply believe that it will not help the readers in general. Figure 2 has some merit, but only to show the well known fact that most normal cells do not express oct4 and some transformed cells including some CSCs cells do (oct4 is an oncogene, an only in some cases correlate with stemness). The real assay for stemness has to be functional clonal growth assays.  It would have made more sense if a figure showed the specific correlation between Oct4 and connexin down/dys-regulation, although I believe this does not really occur (I await the evidence as it would be interesting).

Many “corrections” to my questions has not actually fully addressed the real problem (e,g, simply adding a reference etc), and in some cases the author states it has been corrected yet this does not seem to be reflected at all in the manuscript. An example: In point 6. The author claims Line 118 has been modified and corrected. I see no such change. To state the original problem: Line 118 “Today in metazoans, there exists 20 connexin genes” This remains unchanged. To reiterate: not all metazoans express connexins,only chordate animals do. Moreover the number of genes differ between species, e.g. humans are thought to express 21 connexin genes in up-to-date databases. It might be a simple mistake, but it is my general feeling that a more thorough correction would be required for a journal of this impact.

Critically, the author provides no new evidence to properly support the idea to the key hypothesis that cancer stem cells do not express functional connexins. I have still to see any solid evidence for this in the papers quoted by the author. The author disagrees with papers in extremely respected journals showing CSCs express connexins. This is fine as the field of CSCs is complicated. However, the author  disagrees with publications in numerous highly respected journals that ES cells express GJs or have functional GJIC….claiming “if they had GJIC they would be contact inhibited or differentiate”. This is perplexing, and just to counter on a few key points claimed in the rebuttal letter: a) ES stems cells DO INDEED differentiate if they grow to confluence (that's why they need to be passaged extremely frequently, that's why they spontaneously differentiate extremely easily too). b) Secondly, ES do not need a feeder layer to grow. Classically yes, but many new approaches allow for “feeder-free” growth of these cells. c) Thirdly, with this in mind it is impossible to refute the huge amount of evidence that support ES cells have functional GJIC, certainly when the counterargument is so weak. Just a few examples for interest you may wish to look at (and that I fail to see how can not be acknowledged by any scientist, no matter if having 40 years or 4 months experience):

Study of Gap Junctions in Human Embryonic Stem Cells.

Quote: “GJIC is required for mouse embryonic stem cell maintenance and proliferation
Methods Mol Biol. 2016;1307:105-21. doi: 10.1007/7651_2014_83.

Connexin43 is required for the maintenance of multipotency in skin-derived stem cells.
Stem Cells Dev. 2014 Jul 15;23(14):1636-46. doi: 10.1089/scd.2013.0459.

Feeder-free monolayer cultures of human embryonic stem cells express an epithelial plasma membrane protein profile.
Quote: we showed that hESCs under these conditions primarily express proteins belonging to epithelium-related cell-cell adhesion complexes, including adherens junctions, tight junctions, desmosomes, and gap junctions.
Stem Cells. 2008 Nov;26(11):2777-81. doi: 10.1634/stemcells.2008-0365.

Enhanced generation of human induced pluripotent stem cells by ectopic expression of Connexin 45.

Quote: “human induced pluripotent stem cells (hiPSCs) contained functional gap junctions partially contributed by Connexin 45 (CX45).”
Sci Rep. 2017 Mar 28;7(1):458. doi: 10.1038/s41598-017-00523-y.

Connexin30.3 is expressed in mouse embryonic stem cells and is responsive to leukemia inhibitory factor.
Sci Rep. 2017 Feb 13;7:42403. doi: 10.1038/srep42403

Abrogation of Gap Junctional Communication in ES Cells Results in a Disruption of Primitive Endoderm Formation in Embryoid Bodies.
Stem Cells. 2017 Apr;35(4):859-871. doi: 10.1002/stem.2545. Epub 2016 Dec 20.

Evidence for bystander signalling between human trophoblast cells and human embryonic stem cells.
Sci Rep. 2015 Jul 14;5:11694. doi: 10.1038/srep11694.

Connexin 43 is involved in the generation of human-induced pluripotent stem cells.
Hum Mol Genet. 2013 Jun 1;22(11):2221-33. doi: 10.1093/hmg/ddt074.

Author Response

To begin, my response to Reviewer #3 , I must congratulate him/her for the consistent and logical criticisms to both the original and revised manuscript. Given that I believe he/she accepts the general findings of the ES Oct4 and Connexin data in all the published papers, there is no way that he/she will accept the hypothesis I have presented in this “Concept “ paper, namely, that there exists in all human cancers two kinds of “cancer stem cells”. If I were to accept all of the suggestions made, this “Concept paper” would be his/her concept paper and, in fact, it would no longer be a “concept” paper.

The Reviewer #3’s second review demonstrates that he/she totally disagrees with almost everything about this paper, in spite of stating: “I am very open for new perspectives and especially those that challenge the current view. However, I also believe that convincing evidence needs to be clearly demonstrated, together with a balanced and open-minded view of literature having an “alternative” view. In my view the current perspective is unbalanced in this respect.”  That statement is his/her opinion. To be “open minded" does not, necessarily, mean I must accept all the suggested changes, especially since there are many technical, methodological, and interpretive problems with each of those recommended challenges to our findings with adult human organ-specific stem cells. None of those cited ES papers controlled for the various Oct4 isoforms or pseudogenes. None compared their techniques on both the ES and any adult stem cells at the same time. None had measured Cx expression with functional GJIC on the same sample  as we did.  Even more serious, while Reviewer #3 accepted the conclusions of all these cited ES papers, he/she apparently did not critically examine our papers, showing what we concluded, namely, we always saw, in our normal human adult stem cells, OCT4A in the nuclei of our cells when there was no expression of any Cx or function of gap junctions.

On the other hand, we could find two kinds of human and canine  “cancer stem cells” , those that expressed Oct4A but not Cx’s or functional GJIC and those that did not express OCT4A , but did express Cx  but that did not have functional GJIC.  This reviewer offered no explanation for our reproducible findings. That is what makes our “Concept” paper novel. It is not based on artifacts or fancy. It is obvious that  I have not,  nor would not,  convinced him/her that I do have an open mind. I should inform this reviewer that, in the beginning, I did not have my current view. It was our pioneering discovery in 1987, when we were the  first to have isolated human adult organ-specific stem cells….ten years before the ES discovery by the Wisconsin & Johns Hopkins groups, that my current view, which I have described here, emerged.

Reviewer #3 has demonstrated a real fundamental difference in the role of evolution has played in cancer. Single cell organisms do not develop cancer…,true. However, they did give rise to multi-cellularity. To do that, new genes, functions and phenotypes arose that made it possible for carcinogenesis in multicellular organisms, such as the human being, to occur…stem cells; symmetrical & asymmetrical cell division; low oxygen niches for stem cells; connexin genes & intercellular communication, Oct4 gene & isoforms & pseudogenes; drug resistance, and  the metabolism of glucose from glycolysis to oxidative phosphosphylation.  Comparing the drug-resistant single cell organism to the drug-resistant “cancer stem cell”, one should clearly see the evolutionary link between the two and the reason I feel the Figure one is critical. Has anyone in the biological literature ever made this link? I think not. His/her inability to see the relevance of Figure 1 demonstrates the major philosophical difference we have in the role of evolution in carcinogenesis. I have already discussed this point in several papers and it would take a total re-write of this paper to demonstrate this connection [   Trosko, J.E., “The gap junction as a ‘Biological Rosetta Stone’: Implications of evolution, stem cells to homeostatic regulation of health and disease in the Barker Hypothesis”. J.Cell Commun & Signaling, 5; 53-66, 2011;  Trosko, J.E. and Kang, K.-S., “Evolution of energy metabolism, stem cells and cancer stem  cells: How the Warburg and Barker hypotheses might be linked. Internatl. J. Stem Cells 5: 39-56, 2012; Trosko, J.E., A conceptual integration of extra-, intra-, and Gap junctional inter- Cellular communication in the evolution of multi-cellularity and stem cells: How disrupted cell-cell communication during development can affect diseases later in life. Internatl. J.  Stem Cell Research & Therapy 3: 1-6, 2016; ISSN: 2469-570X; Trosko, J.E. Evolution of microbial quorum sensing to human global quorum sensing: An insight to how gap junctional intercellular communication might be linked to global metabolic disease crisis. Biology 5, X, 2016; doi:10.3390/Special issue: Beyond the modern evolutionary synthesis- What have we missed? ]

 The reviewer, I believe, would agree with me that, from a normal cell in the human body, to ultimately become a cancer involves classic evolutionary changes. However, I view cancer in much broader terms, i.e., that  evolutionary changes, from the single cell organ to the multi-cellular human, made human cancer possible. Having said that, if the Editor and reviewer demand that either I eliminate this approach or add many more pages and references, I will withdraw this manuscript.  To write this manuscript to be the Reviewer’s point of view of a novel “Concept” paper is not my intent. If I am wrong, the scientific community will point this out. If I am correct, and it is not published, the scientific community loses an opportunity to re-focus on new strategies for cancer prevention & treatments.

As to Figure 2, the reviewer #3 states: “Figure 2 has some merit, but only to show the well -known fact that most normal cells do not express oct4 and some transformed cells including some CSCs cells do (oct4 is an oncogene, an only in some cases correlate with stemness).” That “well-know fact” is not a fact with me and my co-workers and collaborators, nor with the giants in the Oct4 field.  Also, reviewer # 3 stated: “The real assay for stemness has to be functional clonal growth assays.  It would have made more sense if a figure showed the specific correlation between Oct4 and connexin down/dys-regulation, although I believe this does not really occur (I await the evidence as it would be interesting).” I suggest reviewer read the paper I cited [  Zhou, Y., Chen, X., Kang, B., She, S., Zhang, X., et al., (2018). Endogenous authentic OCT4A proteins directly regulate FOS/AP-1 transcription in somatic cancer cells, Cell Death & Disease 9: 585; DOI 10.1038/s41419-018-0606-x.]. Also, clonal assays of non-stem cells, such as normal life span-limited fibroblasts or epithelial cells do not express Oct4 genes. However, we have many clones of normal human organ-specific stem cells, in which Oct4 is expressed. The moment we induce differentiation of these Oct4A -positive cells, they stop expressing Oct4A and they divide for a limited amount of time as differentiated progenitor cells. Reviewer should carefully read and look at figures in our papers, such as the Tai et al , Carcinogenesis, 2005 paper.  The role of Oct4A is as a transcription factor for stemness, not for proliferation, but for symmetrical cell division.

In addition, “To state the original problem: Line 118 “Today in metazoans, there exists 20 connexin genes” This remains unchanged. To reiterate: not all metazoans express connexins, only chordate animals do. Moreover the number of genes differ between species, e.g. humans are thought to express 21 connexin genes in up-to-date databases. I am aware of that claim. However, I was at a gap junction meeting where those involved in these studies still had not resolved the issue. I have made the correction to change to reflect only cordates.

Given that, he/she really wants me to totally re-write this manuscript. Since there are several statements he/she made in this 2nd review that leads me to believe he /she does not believe our multiple peer- reviewed publications with human organ-specific adult stem cells ( kidney, breast, pancreas, liver, etc ) that these cells DO NOT have expressed connexins nor functional gap junctions, such as the latest of our human liver adult stem cells [ Chang, C.C., Tsai, J.L., Kuo,K.K., Wang,K.H., Chiang, C.H., Kao, A.P., Tai, M.H., Trosko, J.E. (2004) Expression of oct-4, alpha fetoprotein and vimentin, and lack of gap junctional intercellular communication as common phenotypes for human adult liver stem cells and hepatoma cells. Proc. Am. Assoc. Cancer Res., 45, 642. Paper is currently being prepared.]     I know of all those papers with ES cells and have even been at meetings where I discussed these “discrepancies” with investigators. The real problem is the differences, by which those ES studies with Oct4A and Cx’s were done,  compared to what we have done.  In addition, I found it strange that this reviewer called Oct4 gene an “ oncogene”. . It could only be viewed that way when ES cells, expressing Oct4 are injected by into a n adult animal and it forms a teratoma ( the operational definition of an ES or "iPS" cell, An adult stem cells with expressed Oct4A , when injected into an adult animal do not form teratomas.

“…however, the author disagrees with publications in numerous highly respected journals that ES cells express GJs or have functional GJIC….claiming “if they had GJIC they would be contact inhibited or differentiate”. This is perplexing, and just to counter on a few key points claimed in the rebuttal letter: a) ES stems cells DO INDEED differentiate if they grow to confluence (that's why they need to be passaged extremely frequently, that's why they spontaneously differentiate extremely easily too).  Yes, I agree, since our adult human stem cells will ultimately differentiate when these clones grow too large sizes ( either in 2-D or 3-D cultures). However, it is when these stem cells stop expressing Oct4A and start to express Cx’s. and have functional GJIC that they differentiate. Please look at our paper by Tai et al, Carcinogenesis, 2005. b) Secondly, ES do not need a feeder layer to grow. Classically, yes, but many new approaches allow for “feeder-free” growth of these cells.  I agree. But those feeder layer free media are preforming the functions of the feeder layer, namely, they provide growth factors to stimulate symmetrical cell division and to prevent expression of the Cx’s.  c) Thirdly, with this in mind it is impossible to refute the huge amount of evidence that support ES cells have functional GJIC, certainly when the counterargument is so weak. Just a few examples for interest you may wish to look at (and that I fail to see how cannot be acknowledged by any scientist, no matter if having 40 years or 4 months experience”. It is not impossible, because we have evidence that contradict those papers with our observations and solid data.

Finally, as a Reviewer, you have the right to suggest to the Editor that my manuscript should not be published. If the Editor agrees with you, I will withdraw my manuscript and submit it to another journal.